# The Role of *Akkermansia muciniphila* on Improving Gut and Metabolic Health Modulation: A Meta-Analysis of Preclinical Mouse Model Studies

**DOI:** 10.3390/microorganisms12081627

**Published:** 2024-08-09

**Authors:** Leila Khalili, Gwoncheol Park, Ravinder Nagpal, Gloria Salazar

**Affiliations:** 1Department of Health, Nutrition and Food Sciences, Florida State University, Tallahassee, FL 32306, USA; lk21i@fsu.edu (L.K.); gp21p@fsu.edu (G.P.); rnagpal@fsu.edu (R.N.); 2Center for Advancing Exercise and Nutrition Research on Aging (CAENRA), Florida State University, Tallahassee, FL 32306, USA

**Keywords:** *Akkermansia muciniphila*, *Desulfovibrio*, *Family XIII AD3011* group, *Candidatus Saccharimonas*, gut, metabolic health, inflammation

## Abstract

*Akkermansia muciniphila* (*A. muciniphila*) and its derivatives, including extracellular vesicles (EVs) and outer membrane proteins, are recognized for enhancing intestinal balance and metabolic health. However, the mechanisms of *Akkermansia muciniphila*’s action and its effects on the microbiome are not well understood. In this study, we examined the influence of *A. muciniphila* and its derivatives on gastrointestinal (GI) and metabolic disorders through a meta-analysis of studies conducted on mouse models. A total of 39 eligible studies were identified through targeted searches on PubMed, Web of Science, Science Direct, and Embase until May 2024. *A. muciniphila* (alive or heat-killed) and its derivatives positively affected systemic and gut inflammation, liver enzyme level, glycemic response, and lipid profiles. The intervention increased the expression of tight-junction proteins in the gut, improving gut permeability in mouse models of GI and metabolic disorders. Regarding body weight, *A. muciniphila* and its derivatives prevented weight loss in animals with GI disorders while reducing body weight in mice with metabolic disorders. Sub-group analysis indicated that live bacteria had a more substantial effect on most analyzed biomarkers. Gut microbiome analysis using live *A. muciniphila* identified a co-occurrence cluster, including *Desulfovibrio*, *Family XIII AD3011* group, and *Candidatus Saccharimonas*. Thus, enhancing the intestinal abundance of *A. muciniphila* and its gut microbial clusters may provide more robust health benefits for cardiometabolic, and age-related diseases compared with *A. muciniphila* alone. The mechanistic insight elucidated here will pave the way for further exploration and potential translational applications in human health.

## 1. Introduction

The human gut harbors a diverse microbial community, comprising bacteria, phages, viruses, protists, worms, and fungi, vital for maintaining intestinal functions [1,2]. The gut microbiome influences host metabolism, the immune system, the central nervous system, and hormonal pathways, ultimately impacting immune, metabolic, and brain health [3,4]. Usually, the composition of the gut microbiome remains relatively constant; however, abnormal alterations (‘dysbiosis’) in microbial diversity, composition, and function (microbial-derived metabolites) can predispose the host to a range of disorders, including obesity, type 2 diabetes (T2D), metabolic syndrome, hypertension, cardiovascular disease, inflammatory bowel diseases (IBD), autoimmune diseases, and allergies. Modulating the gut microbiome composition and diversity to restore homeostasis can ameliorate these disorders [5]. Several factors can affect gut microbiome composition, including host genetics, diet, age, mode of birth, and antibiotics. Among these contributing factors, diet has the most profound effect on the regulation and modulation of the gut microbiome [5,6].

Bacteria represent the primary and predominant occupants among the diverse constituents of the gut microbiome, actively engaging in physiological processes through intermediate metabolites or surface antigens [7,8]. *Akkermansia muciniphila* (from now on referred to as *A. muciniphila*) is an essential member of the human as well as rodent gut bacterial microbiome that is inversely associated with body weight, lipid levels, and aging of the host [4].

*A. muciniphila* is an anaerobic Gram-negative bacterium initially identified in human feces. This bacterium utilizes mucin as a source of carbon, nitrogen, and energy [9]. *A. muciniphila* is among the most prevalent bacterial species in the human intestinal microbiota, comprising approximately 0.5% to 5% of the total bacterial population [9,10,11]. Since its initial discovery in 2004, extensive research has been conducted to explore the role of *A. muciniphila* in human metabolic health and disease treatment. In healthy adults, *A. muciniphila* is positively associated with gut and metabolic health [12], whereas an unusual decrease in the *A. muciniphila* population can disrupt the integrity of the gut epithelial barrier, resulting in impaired gut barrier permeability (‘gut leakiness’), elevated plasma endotoxin levels (endotoxemia), aberrant inflammatory responses, and metabolic irregularities [4]. *A. muciniphila* plays a significant role in the pathophysiology of conditions such as obesity, diabetes, inflammatory bowel disease, psychiatric disorders, aging, and other diseases. Numerous studies have highlighted the therapeutic potential of *A. muciniphila* in addressing metabolic disorders [13,14,15] and immune-related diseases [16]. *A. muciniphila* also reduced insulin resistance, inflammation, and adiposity in high-fat diet (HFD)-treated mice, suggesting the beneficial effect of this bacteria in reversing metabolic disorders [17]. The most effective method to increase *A. muciniphila* abundance is dietary supplements such as polyphenols, alkaloids, capsaicin, plant-derived carbohydrates, Chinese medicines, and specific nutritional patterns [18]. Our recent studies demonstrate that consumption of blackberry and gallic acid, a polyphenol enriched in blackberry, ameliorated atherosclerosis [19,20]. Gallic acid fostered *A. muciniphila* abundance in ApoE^−/−^ male mice treated with HFD [19,20]. Interestingly, in female mice, gallic acid did not affect plaque or *A. muciniphila* abundance.

The reduction in inflammatory responses by *A. muciniphila* is a major contributor factor to the health benefits of this bacterium. Inflammation represents a well-coordinated reaction to harmful triggers, aiming to restore the system to its usual state. Obesity, T2D, insulin resistance, and metabolic syndrome are intimately connected to persistent ‘inflammation’, marked by abnormal cytokine generation, heightened production of acute-phase substances and other agents, and the activation of an array of inflammation-related pathways. Definitive evidence from experiments, epidemiological studies, and clinical observations over the past decade establishes a causal relationship between inflammation, including the molecules and networks crucial to inflammatory responses, and the emergence of these metabolic conditions. This is especially notable in the context of metabolic disorders, giving rise to either these ailments or the subsequent complications stemming from inflammatory responses [21]. On the other hand, inflammation can bring about alterations in gut functionality that endure far beyond the point when the initial inflammation has subsided. Effects like harm to enteric nerves, the promotion of pain pathways, and lingering low-grade inflammation might all persist unnoticed through traditional examinations, ultimately contributing to the identification of various GI disorders [22].

In addition to reducing inflammation, several mechanisms of action that may contribute to the beneficial effects of *A. muciniphila*, include improved regulation of gut barrier function, the expression of tight-junction proteins, mucin secretion, and epithelial cell proliferation [13]. Owing to these positive health outcomes and mechanisms, this bacterium is envisaged to hold substantial potential as a next-generation probiotic treatment and therapeutic target. Most evidence of *A. muciniphila* and its health benefits comes from preclinical rodent model studies conducted under divergent experimental, mechanistic, and geographical settings. Therefore, there is a need to systematically integrate and compare these disparate studies to identify reproducible effects and mechanisms associated with *A. muciniphila* effects. To fill this knowledge gap, we herein conduct a meta-analysis and comprehensive scrutiny of the role of *A. muciniphila* in the management of GI and metabolic disorders in mouse models.

## 2. Materials and Methods

This study followed the Preferred Reporting Items for Systematic Reviews and Meta-Analyses (PRISMA) guidelines. The study protocol was registered prospectively in PROSPERO (CRD42023412714).

### 2.1. Search Strategy

A comprehensive search was conducted from January 2000 until May 2024 across several databases, including PubMed, Web of Science, Science Direct, and Embase. The search aimed to identify studies focusing on the impact of *A. muciniphila* supplementation on gut microbiota and metabolic and GI disorders in mice. The search terms encompassed various relevant terms, including (*Akkermansia* OR *Akkermansia muciniphila* OR *Akk*) AND (Aging OR Atherosclerosis OR Diabetes OR Fatty liver disease OR Dyslipidemia OR Obesity OR Inflammation OR Gastrointestinal disorder OR Gut inflammation OR Colitis) AND (Gut microbiome OR Gut microbiota) AND (Mice OR Mouse). Only studies published in the English language were considered for inclusion.

### 2.2. Inclusion and Exclusion Criteria

The inclusion criteria consisted of two main components. Firstly, the chosen studies had to be conducted on mouse models with metabolic or GI disorders. Secondly, these studies needed to examine the impact of *A. muciniphila* supplementation on gut microbial composition and metabolic responses, including the lipid profile, glucose metabolism, and inflammatory markers (circulation and/or gut). Studies supplementing the mice with the live, or heat-killed bacteria as well as proteins or EVs isolated from *A. muciniphila* were also included. On the other hand, we excluded studies with insufficient information regarding experimental details or findings. Additionally, narrative review studies, meta-analyses, and any other articles that did not meet the requirement of being original research studies were also excluded.

### 2.3. Characteristics of Included Studies

The initial exploration across various databases produced 26,734 articles. After eliminating duplicates, books, reviews, randomized control and clinical trials, and meta-analyses, 88 articles were evaluated for potential inclusion. Among these, 48 manuscripts investigated the impact of *A. muciniphila* treatment on microbiome composition, GI health, and inflammatory and metabolic responses in mice. After the titles and abstracts were carefully assessed, nine studies were removed due to insufficient data, and 39 studies meeting the inclusion criteria were selected for the meta-analysis (Figure 1). Among the included studies, the effect of *A. muciniphila* was studied on aging (three studies) [23,24,25], GI disorders (14 studies) [26,27,28,29,30,31,32,33,34,35,36,37,38,39], metabolic disorders (16 studies) [17,40,41,42,43,44,45,46,47,48,49,50,51,52,53,54], and HFD-feeding (six studies) [15,55,56,57,58,59].

The characteristics of the studies, including mice and *A. muciniphila* strains, age, sex, treatment, and overall effects, are shown in Appendix A. Most of the studies used wild-type C57Bl/6 mice (33 studies), while others used ApoE^−/−^, ApoE*3-Leiden, LDLR^−/−^, ICR-CD1, and the accelerated aging model *Ercc1^−/^^Δ7^*(1 study each). Thirty-one studies used males, six used females, one used both males and females, and one study did not report the strain or the sex of the mouse. Regarding the *A. muciniphila* strain, most of the studies used the ATCC BAA-835/CIP107961/CCUG64013 strain (31 studies), one study tested three strains (FSDLZ20M4, FSDLZ36M5, and FSDLZ39M14) [31], one study used the murine strain 139 [33], and one used the KCTC 14172BP strain [34] from human feces. Most of the studies used live bacteria, while some studies compared the effect of the live bacteria with heat-inactivated killed bacteria, protein extracts, or EVs. For statistical analysis, the impact of the live bacteria was compared with all non-alive treatments grouped, which includes heat-killed, protein extracts, and EVs.

### 2.4. Statistical Analyses

The data analyses were conducted utilizing STATA18 (StataCorp, College Station, TX, USA). We followed the PRISMA guideline, as this platform is widely recognized and recommended for systematic reviews and meta-analyses [60]. We employed a restricted maximum likelihood approach for random effects to conduct the meta-analyses. Considering the possibility of inaccessible, unidentified, or unregistered studies, we opted for the random-effect model. We assessed the heterogeneity among the studies by employing the Cochran I-squared, tau-squared, and Q tests. Heterogeneity was considered substantial if the I-squared value exceeded 75% [61]. If significant heterogeneity existed between studies, effect sizes with unacceptable CIs were excluded from the analysis. To report the combined effect size of each study, we utilized the SMD (standardized mean difference) along with its corresponding 95% CI (confidence interval). We employed funnel plots to evaluate potential publication bias and conducted Egger’s and Begg’s tests [62,63].

### 2.5. Data Set Collection and Microbiome Analysis

We identified six publicly available sequencing datasets with accompanying metadata from the studies included in this paper. Raw sequencing data were obtained from the National Center for Biotechnology Information (NCBI) Sequence Read Archive (SRA). One study did not specify the sequencing region, while all other studies amplified the 16S rRNA V3–V4 region. One study did not provide this information regarding the sequencing platform, four studies used the Illumina MiSeq platform, and one used the Ion S5TM XL platform for sequencing. We utilized 35 CTL and 41 *A. muciniphila* samples for microbiome analysis. The sequencing data were processed using QIIME2 (ver. 2-2022.11) [64]. Raw sequences were demultiplexed and filtered based on their quality (Q > 20) using the q2-demux plugin, trimming, and denoising using DADA2 (version 1.26.0) [65]. All identified amplicon sequence variants (ASVs) were aligned using MAFFT (version 7.520) [66]. Taxonomy assignment for the ASVs was performed using the sklearn classifier with the pre-trained naïve Bayes taxonomy classifier, aligned against the 99% SILVA 138 database. To minimize the potential bias introduced by taxa specific to individual studies on overall results, we only considered genera with a prevalence of more than 20% among all samples. Alpha diversity measures were employed to assess microbial diversity, including Chao1 (richness) and the Shannon index (richness and evenness). The non-parametric Kruskal–Wallis test was applied to identify significant differences in microbial diversity. Metagenomic functional activities were predicted using the open-source bioinformatics tool Phylogenetic Investigation of Communities by Reconstruction of Unobserved States 2 (PICRUSt2 (version 2.4.1)) [67]. Sequences were input into PICRUSt2 to predict the functional genes of classified members within the gut microbiota. The inferred gene families were subsequently annotated against Kyoto Encyclopedia of Genes and Genomes (KEGG) orthologs and grouped into KEGG pathways to generate functional pathways. Differential abundance of taxa and predicted functional pathways were identified using the ANOVA-Like Differential Expression (ALDEx2 (version 1.36.0)) approach [68]. Visualization was carried out using Python (version 3.8.16) packages.

## 3. Results

### 3.1. A. muciniphila Reduces Enteric and Systemic Inflammation

Given that inflammation serves as the cornerstone for both the onset and advancement of metabolic and GI disorders, we assessed the impact of *A. muciniphila* on both systemic and gut inflammation.

The common standardized mean difference (SMD) and 95% confidence interval (95% CI) of the serum inflammatory biomarkers are shown in Appendix A, based on a random effect model. Considering the results of the included studies, the overall interventions resulted in significant reductions in TNFα and IL6 expression and a significant upregulation of IL10 in the gut in mice models of metabolic and GI disorders (Figure 2).

In the serum, *A. muciniphila* reduced TNFα and IL6 and showed a non-significant effect in IL10 (trended upwards) and LPS (trended downwards) levels. The results of Egger and Begg tests showed slight to no evidence of publication bias except for serum LPS (Appendix A). Publication bias can be due to the low number of studies for this marker (6 studies).

Sub-group analysis of serum inflammatory markers (Appendix A) showed that both live and non-alive interventions significantly reduced TNFα and IL6. However, only the live bacteria increased IL10 levels. Supplementation for 3–6 weeks was more effective in improving serum inflammatory markers. For the gut, all *A. muciniphila* interventions effectively reduce inflammation. All dose and duration subgroups were effective in lowering TNFα expression. The dose of ≤10^8^ colony-forming units (cfu) and duration of ≤2 weeks caused a significant reduction in IL6 expression, while a dose of ≥10^9^ cfu and a treatment duration of ≤2–6 weeks increased IL10 expression.

### 3.2. A. muciniphila Improves Gut Health Markers

The significant reduction of inflammation in the gut suggests improvement in gut health by *A. muciniphila*. Thus, we analyzed studies measuring colon length, mucus thickness, and the expression of tight-junction proteins in the gut. The SMD and 95% CI of these measurements are shown in Appendix A, based on a random effect model. According to the results of all the studies, the overall interventions resulted in a significant improvement in colon length and mucus thickness (Figure 3) and elevated expression (mRNA and protein) of Zonula occludens 1 (ZO1), occludin, and claudin (claudin-1,3,4, and 8). A non-significant increase was seen in the expression of toll-like receptor-2 (TLR2) and mucin 2 (Muc2) in the gut epithelium, and slight to no evidence of publication bias (Appendix A).

Sub-group analysis (Appendix A) showed that all types of intervention significantly increased colon length, mucus thickness, and the expression of tight-junction proteins. Both dose sub-groups and a duration of ≥7 weeks caused significant improvement in mucus thickness. Both dose sub-groups provided substantial improvement in colon length. All dose and duration sub-groups effectively increased the expression of ZO1 and occludin. Dose of ≥10^9^ cfu and duration of ≤2–6 weeks sub-groups significantly increased claudin expression in the gut.

### 3.3. A. muciniphila Improves Glycemic Control, Lipid Profiles, and Liver Health Markers

Next, we focus on metabolic diseases to assess whether the improvement in gut health was associated with improved lipid and glycemic profiles and markers of liver health. The SMD and 95% CI of the glycemic control factors, including blood glucose, serum insulin level, oral glucose tolerance test (OGTT), and homeostatic model assessment for insulin resistance (HOMA.IR); serum lipids, including triglycerides (TG) and cholesterol; liver health markers including alanine transaminase (ALT) and aspartate aminotransferase (AST) are shown in Appendix A.

Consistent with a reduction in gut inflammation, the effect of the overall intervention with *A. muciniphila* resulted in a significant improvement in glycemic control parameters, lipid profile, and liver enzymes (Figure 4). Publication bias was seen for blood glucose, insulin, and TG (Appendix A).

Sub-group analysis (Appendix A) showed that live *A. muciniphila* reduced blood glucose at a dose of ≥10^9^ cfu and a duration of ≥3 weeks. Both live and non-live *A. muciniphila* reduce total cholesterol and TG. A dose of ≤10^8^ cfu for ≥ 3 weeks reduced total cholesterol level, while a dose of ≥10^9^ cfu for ≤6 weeks caused a significant reduction in TG in mice with metabolic disorders. The results of all subgroup analyses were statistically significant for liver enzymes.

### 3.4. A. muciniphila Improves Body Weight and GI and Metabolic Disorders

The GI disorders studied in the selected studies included dextran sulfate sodium (DSS)-induced colitis, intestinal inflammation caused by reduced IL10, colon perforation, intestinal mucositis, and cadmium-induced intestinal damage, all of which reduced the body weight of the mice. In contrast, animals with metabolic disorders caused by HFD feeding and alcoholic liver disease gained weight.

The overall effect of the interventions improved body weight changes, increasing the weight of mice with GI disorders, and reducing the weight of animals with metabolic disorders (Figure 5 and Appendix A). A slight evidence of publication bias was seen for both parameters (Appendix A).

Sub-group results (Appendix A) showed that all intervention types, doses, and duration reduced the body weight of mice with metabolic disorders. However, only a dose of ≤10^8^ cfu effectively reduced body weight in animals with GI disorders. All studies evaluating the effect of *A. muciniphila* on the body weight of animals with GI disorders utilized live bacteria for ≤2 weeks.

### 3.5. A. muciniphila Remodels the Gut Microbiome

The analysis of *A. muciniphila* studies shows that this bacterium reduces systemic and gut inflammation, improves gut epithelial integrity and glycemic control, and reduces the expression of liver enzymes associated with liver dysfunction. Because of the multiple benefits associated with the increased abundance of *A. muciniphila*, this bacterium likely remodels the microbiome by upregulating health-promoting anti-inflammatory bacteria and/or reducing disease-promoting bacteria. To assess the overall changes in the microbiome induced by *A. muciniphila*, we analyzed six studies with publicly available microbiome data: four studies focused on metabolic disorders, and two centered on age-related conditions (Table 1). Five studies used the same *A. muciniphila* strain; one did not report the strain, and all used the live bacterium.

To assess the impact of *A. muciniphila* treatment on overall gut microbial diversity in these six studies (Table 1), we calculated Shannon and Chao1 alpha-diversity metrics. Although we did not observe a significant difference between the control (CTL) group and the *A. muciniphila*-treated (AKK) group, the Chao1 diversity was marginally lower in the AKK mice (Figure 6A). This suggests that both groups exhibit similar evenness, but the AKK group has less richness. We investigated the average microbial composition at the phylum, family, and genus levels and assessed the differences using the ALDEx2 statistical algorithm.

At the phylum level (Figure 6B), the mouse gut is predominantly dominated by two major phyla, *Bacteroidota* and *Firmicutes*, with no notable differences. As expected, the abundance of the phyla *Verrucomicrobiota* (the bacterial phylum to which the *Akkermansia* genus belongs) was elevated in the AKK group. *Actinobacteriota* was reduced, and *Proteobacteria* was increased in this group. At the family level (Figure 6C), *Akkermansia*, *Muribaculaceaea*, and *Oscillospiraceae* were upregulated, while *Bacteroidaceae* was reduced in the AKK group.

A substantial difference was observed at the genus level in *Akkermansia* abundance between groups (CTL: 0.07%, AKK: 6.11% on average) (Figure 6D). However, the prevalence (i.e., the % detection rate) of *Akkermansia* was almost identical between the groups (CTL: 40.00%, AKK: 46.34%). Additionally, we examined the correlation between *Akkermansia* and other genera. We found that three genera—*Desulfovibrio*, *Family XIII AD3011* group, and *Candidatus Saccharimonas*—showed not only a strong positive correlation with *Akkermansia* abundance but also formed a co-occurrence network with *Akkermansia* exclusively in AKK mice (Figure 6F–H). These genera, however, did not show differential abundance between the CTL and AKK groups. Interestingly, CTL and AKK exhibited distinct microbial co-regulation networks. In the CTL group, there was one major keystone cluster and two sub-clusters of networks. In contrast, in the AKK group, there was one unique keystone cluster and a secondary cluster that combined two sub-clusters, both of which were also identified in the CTL group.

Furthermore, these core clusters shared most of their genera with the clusters in the CTL group. However, only in the AKK group, we observed a subcluster of the network composed of genera that showed a strong positive correlation with *A. muciniphila*. These genera formed major or subtle clusters in the CTL group, indicating that *A. muciniphila* treatment induces changes in the ecological niche and creates a distinct microbial network in the gut (Figure 6G,H). In addition, there was no difference in the abundance of the *Lachnospiraceae* family. Still, several genera in this family, including *Tuzzerella*, *Lachnoclustridium*, and *Lachnospiraceae UCG-001* and *UCG-006*, were more abundant in AKK mice (Figure 6C,E). Meanwhile, *Lactobacillus* decreased in the AKK group and had a mutually exclusive network with several genera, mostly belonging to the *Oscillospiraceae* family, which increased in the AKK group (Figure 6C–E).

Analysis of the three genera in the co-occurrence cluster showed no statistical difference between the control and AKK groups (Figure 7A–C); however, these genera were more abundant in samples in which *A. muciniphila* was observed in both the control and AKK groups (Figure 7D–F). These data suggest that *Desulfovibrio*, *Saccharimonas*, and *Family_XIII_AD3011* have a strong co-occurrence association with *A. muciniphila.*

The PICRUSt-inferred prediction of the microbiome’s functional readout showed an intriguing pattern between CTL and AKK, although it was not statistically significant. The AKK group had an enrichment in pathways associated with steroid biosynthesis (Appendix A). Additionally, functions related to the biosynthesis of carotenoids and flavonoids, known for their antioxidant properties, were more abundant in the AKK group. Notably, some pathways associated with neurodegenerative diseases, including Parkinson’s and Alzheimer’s, were marginally higher in the AKK group. Conversely, pathways related to genetic information processing, such as RNA transport, the sulfur relay system, and ribosome function, were more abundant in the CTL group. Interestingly, the *Desulfovibrio* genera have been associated with Parkinson’s [69], but a direct role in the disease has not been demonstrated.

## 4. Discussion

The present meta-analysis analyzed preclinical studies assessing the effects of *A. muciniphila* (live and heat-killed) and its derivatives, such as EVs and outer membrane protein, on aging and intestinal and metabolic disorders. The results, summarized in Figure 8, show that *A. muciniphila* reduced pro-inflammatory responses in several conditions, including aging, ulcerative colitis, intestinal inflammation, intestinal mucositis, alcoholic liver disease, obesity, HFD-induced non-alcoholic fatty liver disease (NAFLD), and liver injury induced by different stimuli. Overall, pro-inflammatory molecules like IL6 and TNFα were reduced, while the anti-inflammatory cytokine IL10 was upregulated in circulation and the gut. IL10 is secreted by several cell types residing in the gut. For example, commensal intestinal T helper 17 (Th17) cells secrete IL10, which confers these cells an anti-inflammatory phenotype [70]. This phenotype is different from pro-inflammatory Th17 cells, which secrete IL17. Lamina propria macrophages secrete IL10, which acts on T_reg_ cells, driving the expression of forkhead box P3 (Foxp3), which is required for T_reg_ cell’s immune suppressive functions [71]. Lamina propria IL10 also acts in intestinal epithelial cells, reducing gut permeability by regulating the expression of tight-junction proteins [72], increasing mucin secretion from goblet cells [73], and promoting the expansion of stem cells [74].

Regarding inflammatory responses, IL10 reduces the expression of pro-inflammatory cytokines, including TNFα, IL1α, and IL1β [75]. From the studies included in the analysis, only two [25,54] measured IL1β in the gut, showing a significant decrease by the intervention (SMD with 95% CI −17.75 [−30.70, −4.79]. One of the included studies [50] showed a significant reduction in IL1β in the liver. Altogether, this evidence suggests that IL10 mediates the improvements in epithelial barrier function and inflammation induced by *A. muciniphila.*

The reduction in serum cholesterol and the increase in high-density lipoprotein (HDL), which was measured in only one study [59], suggest that *A. muciniphila* upregulates the reverse cholesterol transport pathway. Cholesterol could be excreted in the gut through scavenger receptor class B type 1 (SR-B1; basolateral membrane) and ATP-binding cassette sub-family G member (ABCG) 5/ABCG 8 (apical membrane) transporters, which should increase cholesterol in feces. Alternatively, cholesterol could be eliminated in the liver by forming bile acids. Nian et al. [43], included in this analysis, showed an upregulation of farnesoid X receptor (FXR) in the liver in *A. muciniphila*-treated mice. Activation of FXR inhibits bile acid synthesis in the liver, suggesting that the excretion of cholesterol in feces is the likely mechanism by which *A. muciniphila* reduces serum cholesterol. Increased FXR in the liver may explain the reduction of TG and improved glucose homeostasis. It is also possible that de novo cholesterol synthesis in the liver and/or the gut is also involved. In fact, *A. muciniphila* mucin utilization locus (MUL) genes have been shown to mediate the downregulation of cholesterol synthesis in the colon [76]. It is unknown if cholesterol synthesis could also be reduced in the liver by *A. muciniphila*.

Separate studies have reported that *A. muciniphila* exhibits the potential to mitigate a range of health conditions, including aging-related issues [23,77], inflammation [78], metabolic syndrome [79], diabetes [80], obesity [81], neurodegenerative diseases [82], and adverse effects of cancer therapy [83]. However, despite these promising outcomes, a comprehensive understanding of the intricate molecular mechanisms underlying *A. muciniphila*’s interaction with the host remains incomplete. We identified a co-occurrence cluster (*Desulfovibrio*, *Candidatus Saccharimonas*, and *Family_XIII_AD3011*) that may regulate *A. muciniphila*’s protective effects. Due to the diverse effects of *A. muciniphila* on host metabolism and health, bacteria in this cluster and their metabolites are likely required for *A. muciniphila* effects.

*Desulfovibrio* is a Gram-negative anaerobic bacterium that produces hydrogen sulfide gas (H_2_S). *Desulfovibrio* uses sulfate (cysteine), which can be obtained from the diet or mucin degradation. The fact that both bacteria are found in the mucus layer suggests that *Desulfovibrio* may use the mucin-degrading activity of *A. muciniphila* to obtain sulfate for synthesizing H_2_S.

The role of *Desulfovibrio* in human health has shown conflicting data with evidence providing positive and negative effects. H_2_S at physiological levels promotes cardiovascular health by increasing nuclear factor erythroid 2-related factor 2 (Nrf2) and the antioxidant capacity, leading to reduced LDL oxidation, reducing inflammation (IL6, TNFα, and NF-κB), promoting vasorelaxation, and inhibiting cardiac remodeling [84]. Thus, it is possible that the reduced inflammatory responses seen in *A. muciniphila*-treated mice are mediated by H_2_S.

Overgrowth of *Desulfovibrio* has been demonstrated in bacteremia [85], sepsis [86], and ulcerative colitis [87]. *Desulfovibrio* abundance is also upregulated in patients with Parkinson’s disease [69]. Regarding species, *D. desulfuricans*, *D. fairfieldensis*, and *D. piger*, but not *D. vulgaris*, were highly abundant in feces from patients with Parkinson’s disease compared with healthy controls [88]. Huynh et al. [89] demonstrated that *D. desulfuricans*, *D. fairfieldensis*, and *D. piger* isolated from Parkinson’s patients caused a higher aggregation of α-synuclein and larger aggregates in *C. elegans* compared with the same species isolated from healthy controls.

Zhang et al. [90] identified *D. desulfuricans* as a major bacteria upregulated by HFD in ApoE^−/−^ mice and showed that live but not heat-killed bacteria aggravated HFD-induced atherosclerosis. The effect of *D. desulfuricans* was associated with elevated gut permeability and serum LPS and was reduced by TLR4 inhibition. Interestingly, *A. muciniphila* was reduced by *D. desulfuricans* treatment, which can partially mediate *D. desulfuricans* effects since *A. muciniphila* was shown to reduce plaque and prevent endotoxemia in ApoE^−/−^ mice [54]. These data suggest that the gut microenvironment in Parkinson’s disease and metabolic disorders may promote a pathogenic phenotype in *Desulfovibrio* species that is not seen in the gut of healthy controls since the upregulation of *A. muciniphila* and *Desulfovibrio* reduces inflammation, gut permeability, and endotoxemia. It is also possible that a particular species, like *D. vulgaris*, could be upregulated by *A. muciniphila.* Unfortunately, we could not identify *Desulfovibrio* species from the available microbiome data from the included studies (Appendix A). Also, H_2_S levels were not measured in the included studies. Thus, whether the cluster upregulated by *A. muciniphila* promotes a healthy H_2_S level remains to be elucidated.

The genus *Family XIII AD3011* group is a newly discovered bacterium; thus, little is known about its precise role in host health. Still, this genus has been suggested to be protective against chronic hepatitis B [91], colorectal cancer [92], psoriatic arthritis [93], and hyperuricemia [94]. However, there is a paucity of research on the casual versus causal association of the genus *Family XIII AD3011* group with host intestinal and metabolic health.

Regarding *Candidatus Saccharimonas,* this genus is upregulated by the Mediterranean diet [95], which was associated with improved cognitive function. Further remodeling of the microbiome by fiber, like inulin, ameliorated diabetic nephropathy, increasing *A. muciniphila*, *Candidatus Saccharimonas*, and acetate [96]. *Candidatus Saccharimonas* is a lactate- and acetate-producing bacterium. It has been suggested to modulate the immune response by suppressing TNFα gene expression in macrophages [97], thus playing a protective role in developing allergic disorders in sensitized individuals [98]. Studies have also reported a decreased abundance of *Candidatus Saccharimonas* in animal models of necrotizing pancreatitis and HFD-induced hypertriglyceridemia, indicating its potential anti-inflammatory role [99,100]. Our findings of increased *Candidatus Saccharimonas* in *A. muciniphila* groups suggest a positive correlation between *A. muciniphila* and *Candidatus Saccharimonas* associated with improved inflammatory health through dietary or cross-feeding co-regulation mechanisms.

The exact mechanism by which bacterial overgrowth may promote disease states is unknown. It is possible that *A. muciniphila* may provide a nurturing environment in which *Desulfovibrio* acquires a health-promoting phenotype with reduced H_2_S production. In fact, our analysis shows that *A. muciniphila* upregulated antioxidant pathways, including carotenoids and flavonoid biosynthesis, which may also minimize toxicity mediated by elevated H_2_S. Another protective pathway upregulated by *A. muciniphila* in our analysis is autophagy. Interestingly, the sulfur relay pathway was reduced in the supplemented group, suggesting that available sulfur may be reduced. Measurements of H_2_S are needed to understand the functional interaction of *A. muciniphila* and *Desulfovibrio*.

Changes in bacterial abundance and pathogenic phenotypes could likely result from several factors, like changes in the patient’s diet, infections, systemic inflammation, gut LPS, liver metabolism, and composition of the mucin layer and gut-derived metabolites.

The protective intestinal mucus layer comprises mucins, highly glycosylated proteins that create a barrier against pathogens. The major glycoprotein is Muc2. Glycans, which include most of the mucin mass, can be modified by adding sialic acid, sulfate, and fucose molecules, which confer a substantial negative change to mucins. Thus, the degree of glycan modifications regulates the interaction of the mucus layer with the bacteria. A study in ulcerative colitis patients showed that both *A. muciniphila* and *Desulfovibrio* bind to mucin, showing a higher binding to mucin isolated from the ulcerative colitis patients, compared with healthy controls [101]. Glycan modification is also altered in ulcerative colitis. For example, sulfation [102] and glycosylation [103] were reduced in patients compared with healthy controls. It is unknown whether changes in mucin modifications may increase the interaction of *A. muciniphila* and/or *Desulfovibrio* with the intestinal epithelium promoting a pathogenic phenotype in these bacteria. Overall, the intestinal environment (inflammation, changes in mucin modifications, mucin content) and bacterial diversity, relative abundance and interaction with the intestinal epithelium may contribute to the health benefits of the *A. muciniphila* bacterial cluster.

Regarding the effect of the live and heat-killed bacteria, significance was seen for most of the markers for both treatment types. The lack of significance for the heat-kill treatment can be due to the low number of studies. For example, for serum IL10 (*n* = 2, *p* = 0.22), gut IL6 (*n* = 3, *p* = 0.2), and blood glucose (*n* = 3, *p* = 0.07). For serum TG, heat-killed treatment has enough studies (*n* = 6, *p* = 0.16), but in some of these studies, the dose was not optimum (most of them used ≤10^8^ cfu, which was not effective in reducing TG level).

TLR-2 was the only marker that was not affected by *A. muciniphila* treatment. Although the expression was not altered, several reports have shown that *A. muciniphila* mediates its effects through TLR receptors. For example, *A. muciniphila* reduced inflammation induced by HFD or LPS through the TLR4 signaling pathway [104]. Remarkably, *A muciniphila* in both live and pasteurized forms and the outer membrane protein (Amuc_1100) activate TLR2/4 pathways [104]. This activation contributes to enhanced mucin secretion and the expression of tight-junction proteins, thus reinforcing the intestinal barrier’s functionality. These effects are particularly significant in cases of obesity and metabolic disorders resulting from Western-style diets (WD), HFD, or high-fructose and high-cholesterol (HFHC) content [50,54,105]. Additionally, *A muciniphila* can stimulate thermogenesis by interacting with intercellular adhesion molecule 2 (ICAM2), while its secreted P9 protein fosters glucagon-like peptide 1 (GLP1) secretion by L cells [45]. This interaction contributes to the maintenance of glucose homeostasis. *A muciniphila*’s benefits also extend to suppressing carbohydrate absorption and promoting increased energy expenditure and the turnover of intestinal epithelial cells [55].

Obesity is currently a primary global health concern, accompanied by metabolic disorders such as cardiometabolic complications and insulin resistance (IR) [106]. These metabolic disorders are closely linked to chronic inflammation and alterations in the gut microbiota composition, including changes in the prevalence of *A. muciniphila* [44,106]. Notably, obese individuals exhibit a higher abundance of *Firmicutes*, whereas the presence of *A. muciniphila* is significantly reduced in individuals with obesity and metabolic disorders. *A. muciniphila*’s levels are inversely correlated with body fat mass and glucose intolerance [15,107]. Furthermore, rigorous quantitative analysis in Chinese subjects has demonstrated a strong correlation between *A. muciniphila* abundance, body mass index (BMI), and the use of antidiabetic medications [108].

Research involving animal models of obesity and T2D has revealed a significant decline in the abundance of *A. muciniphila*, which can be restored to normal levels through prebiotic supplementation. This restoration aligns with an improved metabolic profile [107]. EVs derived from *A. muciniphila* have shown the capacity to induce significant body and fat weight reductions in HFD-fed animals [57]. Both *A. muciniphila* and EVs derived from it enhanced the integrity of the intestinal barrier, reduced inflammation, improved energy balance, and positively impacted blood parameters such as lipid profiles and glucose levels in HFD-fed animals.

Although only a handful of studies in humans have used *A. muciniphila* supplementation, the results are promising and align with the overall observations of our meta-analysis. In a proof-of-concept exploratory study in obese individuals, *A. muciniphila* supplementation improved insulin sensitivity and reduced total cholesterol, liver enzymes, inflammation, body weight, and adiposity [14]. This study also showed that oral supplementation with 10^9^ and 10^10^ cfu alive or pasteurized *A. muciniphila* for three months was safe and well tolerated [14].

Regarding short-chain fatty acids (SCFAs), the included studies have limited data on SCFAs. *A. muciniphila* and its metabolic byproduct, propionate, can affect cellular lipid metabolism through the regulation of various factors, including G protein-coupled receptor 43 (GPR43), fasting-induced adipose factor (FIAF), peroxisome proliferator-activated receptor gamma (PPAR), and histone deacetylase (HDACs) [109]. Notably, pasteurization of *A. muciniphila*, in addition to live *A. muciniphila* and its metabolites, has demonstrated comparable effectiveness in mitigating fat mass accumulation, IR, and dyslipidemia in mouse models when compared to the non-pasteurized bacterium [44]. Furthermore, an exciting discovery is the stability of Amuc_1100, a specific protein derived from *A. muciniphila*’s outer membrane, to enhance gut barrier function and partially replicate the beneficial impacts of *A. muciniphila* by engaging with TLR2 [44]. It is worth noting that the genotype of *A. muciniphila* can differentially influence brown adipose tissue inflammation and the whitening process in mice fed HFD [46].

Regarding limitations, only one of the included studies used both sexes reducing the impact of our findings on women’s health. It is unknown if *A. muciniphila* co-occurrence bacterial cluster can be also present in female mice and whether health-promoting benefits are mediated by similar mechanisms. Further, not all studies measured all the markers of interest (e.g., IL1β) or have enough studies testing *A. muciniphila* derivatives (e.g., membrane protein isolates, EVS).

## 5. Conclusions

Taken together, the findings from this study demonstrate that the co-occurrence of *A. muciniphila* with other gut bacterial clades, including *Desulfovibrio*, *Family XIII AD3011* group, and *Candidatus Saccharimonas* confers beneficial effects on host gut and cardiometabolic health through different molecular pathways and mechanisms. For instance, *A. muciniphila* and its protein derivatives can regulate TLR signaling, remodel the mucin layer, and provide sulfur sources for *Desulfovibrio*. *Desulfovibrio* may provide physiological levels of H_2_S, which is beneficial for the cardiovascular system, while *Candidatus Saccharimonas* may provide SCFAs like lactate and acetate. Thus, it is likely that a probiotic consortium containing these four gut bacterial genera would be more effective in a wide range of inflammatory diseases.

## Figures and Tables

**Figure 1 microorganisms-12-01627-f001:**
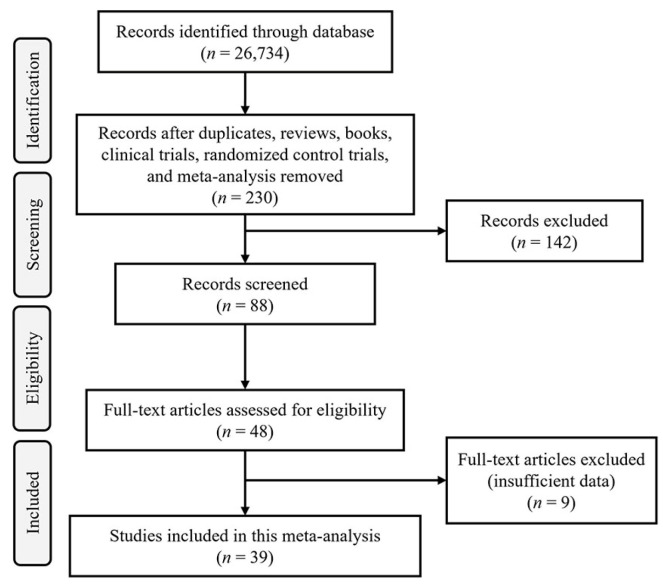
The PRISMA flowchart of the approach employed in this study. The relevant studies were identified through comprehensive database searches in PubMed, Embase, Web of Science, and Science Direct up to May 2024. The search criteria encompassed studies exploring the impact of *A. muciniphila* on gut microbiota and metabolic response, specifically in mice models, which are used to study metabolic and GI disorders.

**Figure 2 microorganisms-12-01627-f002:**
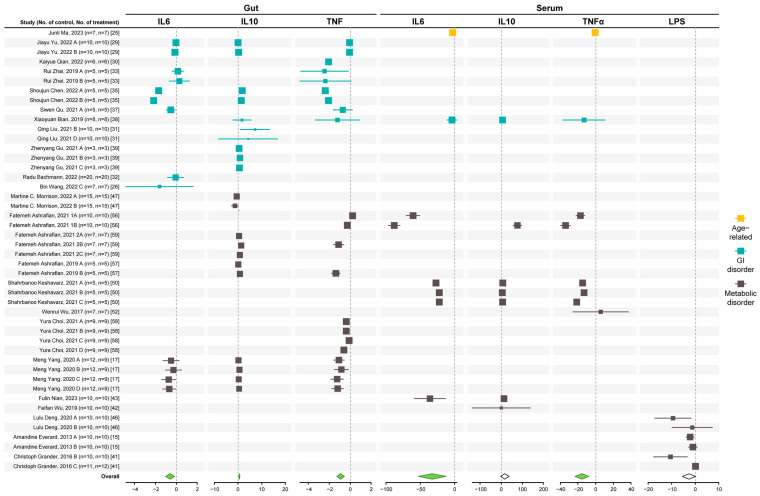
Effect of *A. muciniphila* on systemic and gut inflammation [15,17,25,26,29,30,31,32,33,35,37,38,39,41,42,43,46,47,50,52,56,57,58,59]. Forest plot of individual SMD of serum inflammatory markers (TNFα, IL6, IL10, and LPS) and expression of inflammatory factors (TNFα, IL6, and IL10) in the gut of aging mice and mice with GI and metabolic disorders. A green diamond indicates significance, while the white diamond indicates not statistical differences.

**Figure 3 microorganisms-12-01627-f003:**
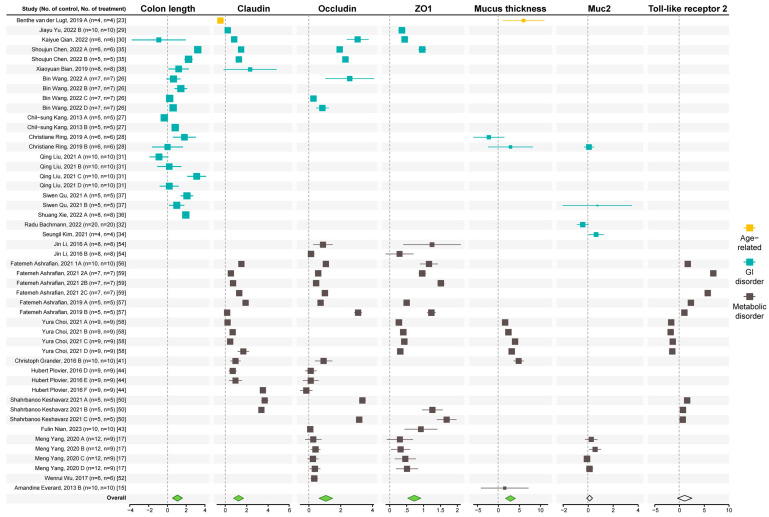
Effect of *A. muciniphila* on gut epithelial health markers [15,17,23,26,27,28,29,30,31,32,34,35,36,37,38,41,43,44,50,52,54,56,57,58,59]. Forest plot of individual SMD of colon length, mucus thickness, and tight-junction expression (protein and mRNA) in the gut of aging mice and mice with GI and metabolic disorders. A green diamond indicates significance, while the white diamond indicates not statistical differences.

**Figure 4 microorganisms-12-01627-f004:**
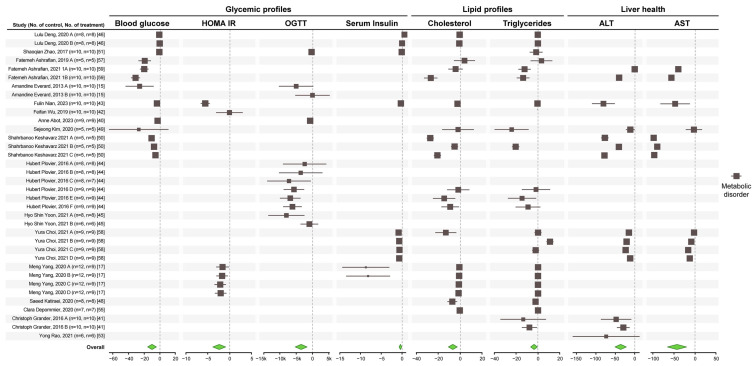
Effect of *A. muciniphila* on metabolic profiles and liver health [15,17,40,41,42,43,44,45,46,48,49,50,51,53,55,57,58,59]. Forest plot of individual SMD of glycemic control (blood glucose, insulin level, and HOMA.IR), lipid profile (TG and cholesterol), and liver enzymes (ALT and AST) of mice with metabolic disorders. A green diamond indicates significance.

**Figure 5 microorganisms-12-01627-f005:**
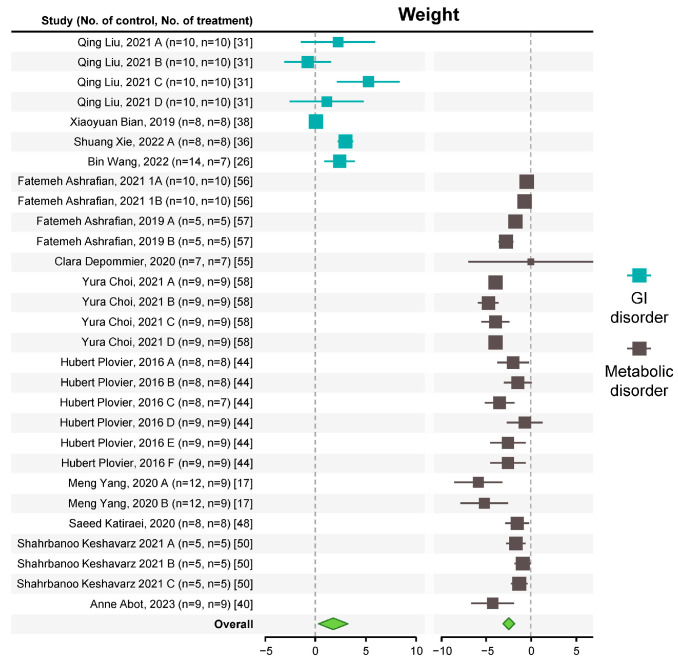
Effect of *A. muciniphila* on body weight [17,26,31,36,38,40,44,48,50,55,56,57,58]. Forest plot of individual SMD of body weight of mice with GI and metabolic disorders. A green diamond indicates significance.

**Figure 6 microorganisms-12-01627-f006:**
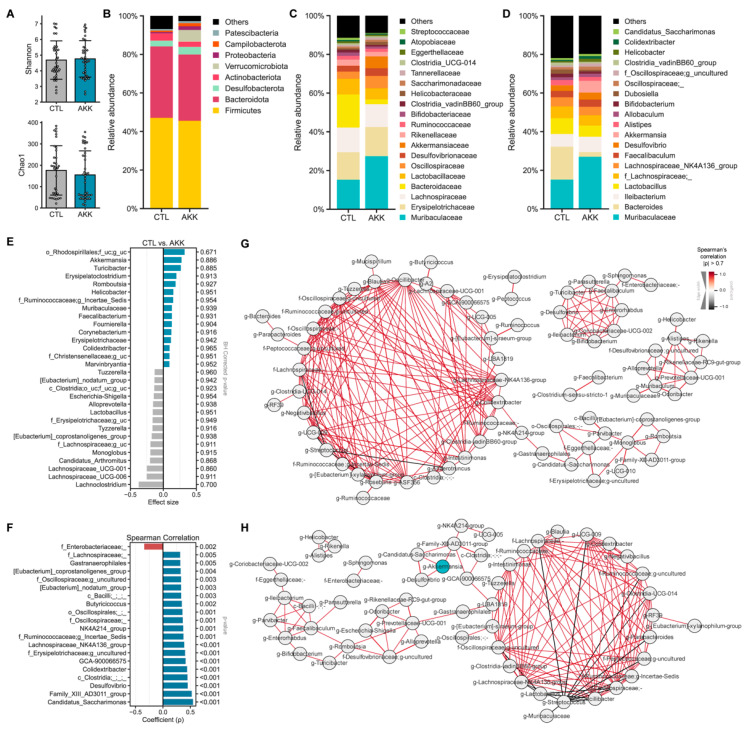
The treatment with *A. muciniphila* induces alterations in microbial co-regulation ecological niches. (**A**) Microbial alpha-diversity. Microbial composition at (**B**) phylum, (**C**) family, and (**D**) genus level. (**E**) The top 15 genera are more abundant in each group (ALDEx2). (**F**) Significantly correlated genus with *A. muciniphila* (Spearman correlation coefficient (ρ) > 0.3). Correlational network between genera in (**G**) CTL and (**H**) AKK group. Each node represents one genus, and only significant links are shown (Spearman coefficient (ρ) > 0.7, Benjamini–Hochberg corrected *p*-value < 0.05). CTL: control group; AKK: *A. muciniphila*-treated group.

**Figure 7 microorganisms-12-01627-f007:**
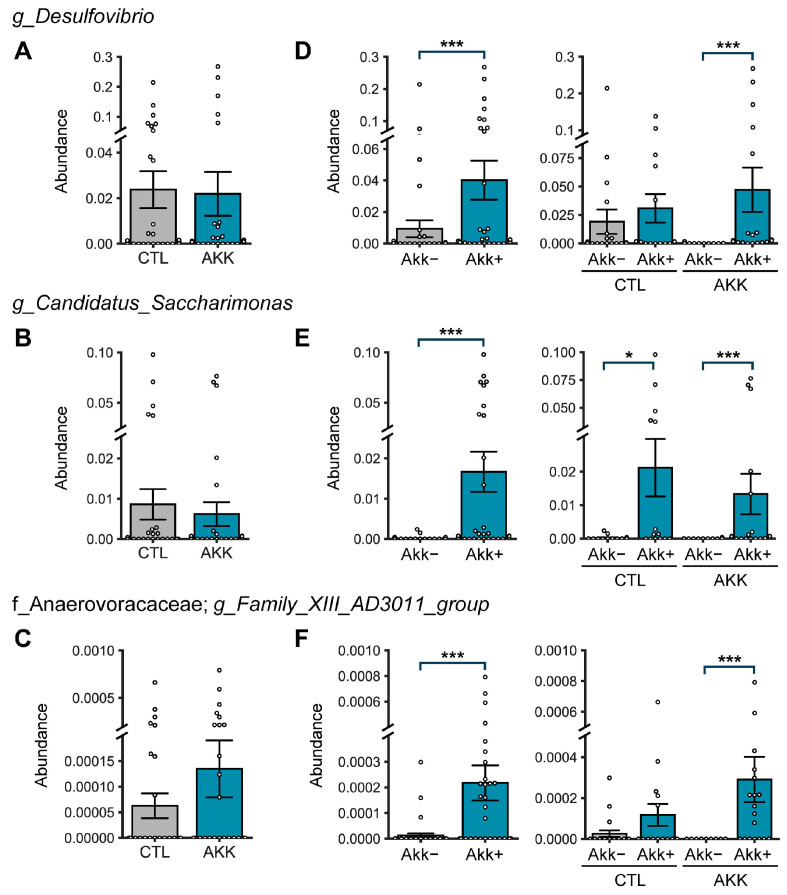
Relative abundance of taxa showing co-occurrence association with *Akkermansia* (Akk) in network analysis of the AKK group. Comparison of the relative abundance of (**A**) *Desulfovibrio*, (**B**) *Candidadus Saccharimonas*, and (**C**) *Family XIII AD3011* group (*Anaerovoracaceae* family) between treatment groups (CTL and AKK), Akk non-detected (Akk-; *n* = 43) and Akk detected (Akk+; *n* = 33) samples. (**D**–**F**) Akk detected and non-detected samples within treatment groups (CTL/Akk− *n* = 21, CTL/Akk+ *n* = 14, AKK/Akk− *n* = 22, AKK/Akk+ *n* = 19). *p*-values were calculated using the Mann–Whitney U test (Wilcoxon rank sum test). Bar plots are presented as mean ± SE. * *p* < 0.05; *** *p* < 0.001.

**Figure 8 microorganisms-12-01627-f008:**
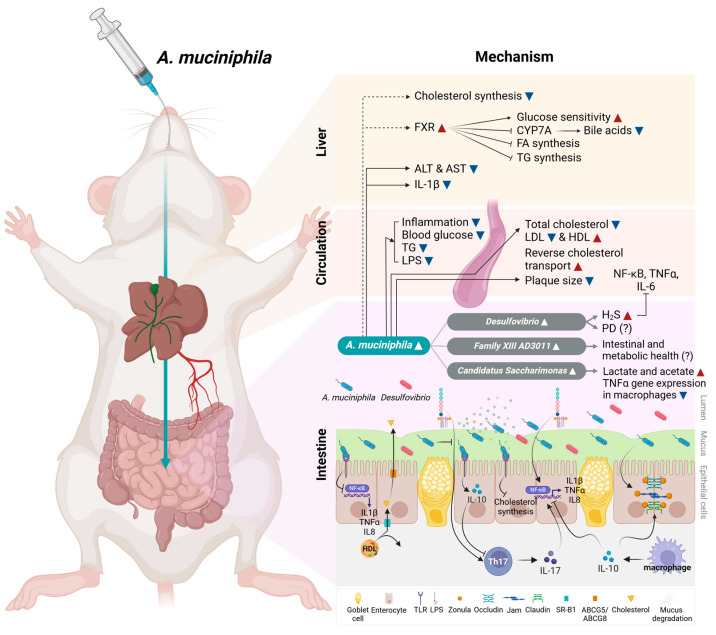
Model of the proposed mechanism by which *A. muciniphila* improves host health. Oral administration of *A. muciniphila* remodels the gut microbiota, increasing the abundance of *Desulfovibrio*, *Candidatus Saccharimonas*, and *Family_XIII_AD3011*. *Desulfovibrio* produces H_2_S, which inhibits inflammation and protects the cardiovascular system. However, H_2_S overproduction has been reported in Parkinson’s disease (PD) patients. *Candidatus Saccharimonas* produces lactate and acetate and reduces inflammation in macrophages. Less is known of the function of *Family_XIII_AD3011*. These changes were associated with reduced intestinal inflammation and improved gut permeability, likely through the activation of TLR and the inhibition of NF-kB. Upregulation of intestinal IL10 likely reduces pro-inflammatory cytokines and upregulates tight junction proteins. In circulation, *A. muciniphila* reduced inflammation (TNFα, IL6), cholesterol, TG, and LPS while increasing HDL and improving glucose control. In the liver, *A. muciniphila*, reduced ALT, AST, and IL1β. The known activation of FXR (dotted line) by *A. muciniphila* may explain the reduction in TG and improved glucose sensitivity. FXR activation, along with the reduction in cholesterol, is also expected to reduce bile acid and fatty acid (FA) synthesis in the liver. Blue and red arrowheads indicate reduced and increased expression, respectively. The model was generated using BioRender.

**Table 1 microorganisms-12-01627-t001:** Included studies for microbiome analysis.

Study	Disease	Treatment	Treatment	Ref.
Xiaoyuan Bian (2019)	GI disorder (Ulcerative Colitis)	Live Akkermansia	CTL (6), AKK (6)	[38]
Seungil Kim (2021)	GI disorder (gut homeostasis)	Live Akkermansia	CTL (4), AKK (4)	[34]
Benthe Van (2019)	GI disorder (Age-related decline in thickness of colonic mucus layer)	Live Akkermansia	CTL (9), AKK (11)	[23]
Rui Zhai (2019)	GI disorder (Chronic Colitis)	Live Akkermansia	CTL (5), AKK (9)	[33]
Fulin Nian (2023)	NAFLD	Live Akkermansia	CTL (6), AKK (6)	[43]
Junli Ma (2023)	Aging-related disorders	Live Akkermansia	CTL (5), AKK (5)	[25]

In total 6 studies (CTL: 35, AKK: 41) were included.

## Data Availability

The original contributions presented in the study are included in the article/Appendix A, further inquiries can be directed to the corresponding author.

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
