# Peer review of "The Role of Akkermansia muciniphila on Improving Gut and Metabolic Health Modulation: A Meta-Analysis of Preclinical Mouse Model Studies"

_microorganisms, 2024, doi:10.3390/microorganisms12081627_

Round 1

Reviewer 1 Report

Comments and Suggestions for Authors

Please consider the following suggestions for revision:

1. Title: The title "Akkermansia muciniphila on gut and metabolic health: a meta-analysis of preclinical mouse model studies" is clear and effectively communicates the essence of the study. However, considering the focus on the microbiome's modulation by A. muciniphila, it might be beneficial to reflect this aspect in the title for enhanced specificity.

2. Abstract: It could be enhanced by including a brief statement on the implications of the findings for future research or clinical applications.

3. Methods: The authors should provide a more detailed criteria used for study selection to ensure transparency and reproducibility of the study selection process.

4. Some of the text in Figure 6 and 8 is too small to read clearly; the font size should be adjusted.

5. Discussion: It would be valuable to include a explicit discussion of the study's limitations.

6. Figure 8 is complex and may benefit from additional explanation in the figure legend.

Comments on the Quality of English Language

Minor editing of English language required

Author Response

We thank the reviewer for the suggestions to improve our manuscript. 

Comment: The title "Akkermansia muciniphila on gut and metabolic health: a meta-analysis of preclinical mouse model studies" is clear and effectively communicates the essence of the study. However, considering the focus on the microbiome's modulation by A. muciniphila, it might be beneficial to reflect this aspect in the title for enhanced specificity

Response: The tile was revised as suggested

Comment: Abstract: It could be enhanced by including a brief statement on the implications of the findings for future research or clinical applications.

Response: The abstract was revised as suggested

Comment: Methods: The authors should provide a more detailed criteria used for study selection to ensure transparency and reproducibility of the study selection process.

Response: the method section was revised.

Comment: Some of the text in Figure 6 and 8 is too small to read clearly; the font size should be adjusted.

Response: The font size was increased as suggested.

Comment: Discussion: It would be valuable to include a explicit discussion of the study's limitations.

Response: Limitations of the study were added, as suggested

Comment: Figure 8 is complex and may benefit from additional explanation in the figure legend.

Response: The figure legend was revised, as suggested

Reviewer 2 Report

Comments and Suggestions for Authors

The authors of the meta-analysis “Akkermansia muciniphila on gut and metabolic health: a meta-analysis of preclinical mouse model studies” have done an extensive review of the literature regarding the effects of Akkermansia muciniphila (A. muciniphilia) on gastrointestinal and metabolic disorders examined in mouse models. The results will not evaluate the translational value of the findings, but an extensive analysis of available data is still valuable and interesting. The included articles are well chosen and the reasons for including them nicely described. The statistical analysis is correctly performed and the figures are of high quality. The included references are pertinent and the conclusions follow the results. However, there are some concerns with the manuscript:

1: The organization of the text is not completely logical. Introduction should be in the introduction section, results should be in the results section and methods should be in the methods section.

A: Line 66-74: Not introduction but clearly results. The same for lines 86-88. The data presented should be in the results section, not the introduction.

B: Lines 89-93 are just a repletion of what you said starting on line 79. Please remove. However, you may want to briefly mention the motive for doing this study (potential translational applications, presumably).

C: Please rewrite section 3.2 and start with the results, that is the findings from the included articles. This is the results section, not an introduction or discussion. Please first answer what the articles found. The findings are discussed in the discussion section. The best option would be to move the first section (line 189-205) to the introduction.

D: Figure 1 should be in the materials and methods section, not the results.

2: The logical reasoning behind sentences.

A: Line 24: “The data suggest that A. muciniphila improves host health via gut microbiome modulation, likely by increasing health-promoting metabolites like hydrogen-sulfide (H2S).

How does gut microbiome modulation result from increased health-promoting metabolites? The connection is not direct. One suggestion is to move this whole discussion to the discussion section, where you already elaborate on this topic.

B: The sentence starting on line 79 should not start with “however”. Please rewrite to something like: “Studies in rodent models have been conducted under divergent experimental, mechanistic and geographical settings, and to compare these different settings we conducted a meta-analysis to investigate the role of A. muciniphila in the management of gastrointestinal (GI) and metabolic disorders in mouse models.” Otherwise, the text is misleading.

C: Please comment on your statement starting on line 517: A study in ulcerative colitis patients showed that both A. muciniphila and Desulfovibrio bind to mucin, showing a higher binding to mucin isolated from the ulcerative colitis patients, compared with healthy controls.

If binding is increased in disease, how does this relate to a beneficial effect of these bacteria?

3: References to included articles and explanation of acronyms.

A: The section starting in line 307: It is not clear where the data comes from. Please indicate in the first lines of the section what studies you used to calculate the diversity and other results presented in this section.

B: Please make sure to explain the acronym EV at first mention in the text.

C: Line 526: What do you mean by the acronym TG?

Author Response

The authors of the meta-analysis “Akkermansia muciniphila on gut and metabolic health: a meta-analysis of preclinical mouse model studies” have done an extensive review of the literature regarding the effects of Akkermansia muciniphila (A. muciniphilia) on gastrointestinal and metabolic disorders examined in mouse models. The results will not evaluate the translational value of the findings, but an extensive analysis of available data is still valuable and interesting. The included articles are well chosen and the reasons for including them nicely described. The statistical analysis is correctly performed and the figures are of high quality. The included references are pertinent and the conclusions follow the results. However, there are some concerns with the manuscript:

1: The organization of the text is not completely logical. Introduction should be in the introduction section, results should be in the results section and methods should be in the methods section.

A: Line 66-74: Not introduction but clearly results. The same for lines 86-88. The data presented should be in the results section, not the introduction.

Response: Sentence in line 66-74 was revised and results in lines 86-88 were removed

B: Lines 89-93 are just a repletion of what you said starting on line 79. Please remove. However, you may want to briefly mention the motive for doing this study (potential translational applications, presumably).

Response: Lines 80-93 were removed.

C: Please rewrite section 3.2 and start with the results, that is the findings from the included articles. This is the results section, not an introduction or discussion. Please first answer what the articles found. The findings are discussed in the discussion section. The best option would be to move the first section (line 189-205) to the introduction.

Response: The section on inflammation was moved to the introduction.

D: Figure 1 should be in the materials and methods section, not the results.

Response: The figure was moved to the Methods section

2: The logical reasoning behind sentences.

A: Line 24: “The data suggest that A. muciniphila improves host health via gut microbiome modulation, likely by increasing health-promoting metabolites like hydrogen-sulfide (H2S).

How does gut microbiome modulation result from increased health-promoting metabolites? The connection is not direct. One suggestion is to move this whole discussion to the discussion section, where you already elaborate on this topic.

Response: The sentence was removed from the abstract.

B: The sentence starting on line 79 should not start with “however”. Please rewrite to something like: “Studies in rodent models have been conducted under divergent experimental, mechanistic and geographical settings, and to compare these different settings we conducted a meta-analysis to investigate the role of A. muciniphila in the management of gastrointestinal (GI) and metabolic disorders in mouse models.” Otherwise, the text is misleading.

Response: The sentence was revised.

C: Please comment on your statement starting on line 517: A study in ulcerative colitis patients showed that both A. muciniphila and Desulfovibrio bind to mucin, showing a higher binding to mucin isolated from the ulcerative colitis patients, compared with healthy controls.

If binding is increased in disease, how does this relate to a beneficial effect of these bacteria?

Response: the sentence was revised to indicate that the intestinal environment may be critical in the regulation of A. muciniphila and Desulfovibrio function.

3: References to included articles and explanation of acronyms.

A: The section starting in line 307: It is not clear where the data comes from. Please indicate in the first lines of the section what studies you used to calculate the diversity and other results presented in this section.

Response: The data was generated from the 6 studies in Table 1, which is now clarified.

B: Please make sure to explain the acronym EV at first mention in the text.

Response: EV is now defined in the abstract.

C: Line 526: What do you mean by the acronym TG?

Response: TG is for triglycerides, which was defined in line 257 of the original submission